# An organic proton cage that is ultra-resistant to hydroxide-promoted degradation

Chase L. Radford[1], Torben Saatkamp[1], Andrew J. Bennet ●[1] ✉ & Steven Holdcroft ●[1] ✉

Alkaline polymer membrane electrochemical energy conversion devices offer the prospect of using non-platinum group catalysts. However, their cationic functionalities are currently not sufficiently stable for vapor-phase applications, such as fuel cells. Herein, we report 1,6-diazabicyclo[4.4.4]tetradecan-1,6-ium (in-DBD), a cationic proton cage, that is orders of magnitude more resistant to hydroxide-promoted degradation than state-of-the-art organic cations under ultra-dry conditions and elevated temperature, and the first organic cation-hydroxide to persist at critically low hydration levels ( < 10% RH at 80 °C). This high stability against hydroxide-promoted degradation is due to the unique combination of endohedral protection and intra-bridgehead hydrogen bonding that prevents the removal of the inter-cavity proton and lowers the susceptibility to Hofmann elimination. We anticipate this discovery will facilitate a step-change in the advancement of materials and electrochemical devices utilizing anion-exchange membranes based on in-DBD that will enable stable operation under extreme alkaline conditions.

One of the greatest challenges to alkaline anion-exchange membrane (AEM) technologies stems from the basic and nucleophilic nature of the hydroxide counter-ions, which readily degrade their cationic organic functional groups[1,2]; degradation increases in severity under dry conditions, as the ratio of water to hydroxide ions (hydration number, $\lambda$) decreases[3–5]. Promising vapor-phase AEM technologies such as high temperature or dry-cathode operated AEM water electrolyzers[6–8], AEM fuel cells[1,9,10], and CO/$CO_2$ electrolyzers[11–15] rely on humidified gas inputs; however, localized regions of substantial dehydration may occur within these cells during operation[16–18]. Drying-out or lowering of the local relative humidity (RH) in these devices increases the alkalinity within the AEM by reducing $\lambda$ and causes rapid degradation of the organic cations used as functional groups[16,19–23]. Similarly, elevated temperature increases the rate of nucleophilic attack on the cationic moieties[5,23,24]. To overcome this key challenge, it is thus imperative to improve the durability of organic cations in highly caustic environments[16,19–23].

The development of more durable AEMs has already benefited significantly from increasing the alkaline stability of cation-hydroxides. Our group, for example, developed steric protection strategies to substantially enhance the stability of (benz)imidazolium cations[23,25–27], while other research groups have enhanced the durability of ammonium cations by steric protection with cyclohexane 'chair' configurations[5,22,28], and by limiting degradative de-alkylation by controlling alkyl-chain lengths[20,23,29]. Despite these advances in cation design, their lifetimes remain insufficient, especially for those applications that are prone to low-humidity conditions and elevated temperatures[2,30,31]. To provide adequate ex-situ assessment of the alkaline stability of organic cations, it is crucial to understand which conditions they are exposed to in the device: during operation of an AEM fuel cell, for example (typically at 40–80 °C), water is consumed at, and electro-osmotically dragged away from, the cathodic catalyst layer[16,32–34]. As a consequence, local electrochemical reaction conditions in these devices can drop to the equivalent of <10% RH[16,34]; a condition where typically $\lambda$-values (water molecules per hydroxide) of 1-2 are reached, akin to hydroxide concentrations of >28 M[21]. In order to achieve established lifetime targets, for example, 5000 h in fuel cells, a significant improvement of the low-$\lambda$ stability of cationic functional groups is urgently needed[31]. It is, therefore, critical to explore organic cation-hydroxides under low RH/elevated

[1]Department of Chemistry, Simon Fraser University, Burnaby, Canada. ✉e-mail: bennet@sfu.ca; holdcrof@sfu.ca

temperature conditions, as resilience under these harsh conditions is the key to unlocking vapor-phase AEM technology.

In this work, the stability of cations that are considered state-of-the-art and are typically incorporated in AEMs to enable hydroxide transport is assessed. Currently, the most hydroxide-stable organic cations are 6-azaspiro[5.5]undecan-6-ium (ASU), 1,3-dibutyl-2-mesityl-4,5-diphenyl-1H-imidazol-3-ium (BMI), and tetrakis(cyclohexyl(methyl)amino)phosphonium (TCAP) (Fig. 1a)[20,22,23]. Although highly stable under caustic conditions when fully hydrated, none are considered stable over extended periods under low RH/elevated temperature applications[1,11,35]. In addition to high stability, cations should possess a low molecular weight to enable the formation of polymers with high ion exchange capacity (IEC) and thus high hydroxide conductivity[36,37]. Below is shown a range of additional cations that have been implemented in AEMs: N,N-dimethylpiperidin-1-ium (DMP), 1,3-dimethyl-2-mesityl-4,5-diphenyl-1H-imidazol-3-ium (MMI), and variations of differently tethered tetramethylammonium (TMA) groups. The latter are synthetically readily accessible and used widely, even though some have a well-documented instability/reactivity to hydroxide such as benzyl trimethylammonium (BTMA)[5,22,23,38], which serves as a baseline in this work.

Our goal was to identify a new approach to produce hydroxide-stable organic cations that are also viable candidates for future incorporation into polymers and AEMs. This current work took inspiration from Alder and coworkers' investigation into unconventional bonding modes[39]. They reported an interesting cationic structure with a proton endohedrally trapped inside the internal cavity of a diamine, namely 1,6-DiazaBicyclo[4.4.4]tetraDecan-1,6-ium (in-DBD, Fig. 1b). Intriguingly, they reported no deprotonation of the internal proton in refluxing NaOD (1 M), NaNH$_2$/NH$_3$ at −33 °C, or upon heating the hydroxide salt to 160 °C for 0.5 h, but the report lacked experimental evidence[39]. Recognizing that in-DBD contains unique bonding modes that differ from common "stable" cations and are formed via a versatile synthetic methodology that could lead to polycations possessing a high IEC, it presented an attractive molecule for study. We, therefore, synthesized in-DBD and found it to possess unprecedented stability in aggressive alkaline media. We found that in-DBD, wherein the cationic charge is carried by a trapped proton, possesses remarkable stability due to the endohedral protection of the proton offered by the aliphatic cage, which is further stabilized between two nitrogen atoms via a symmetrical intra-bridgehead hydrogen bond. In-DBD exhibits orders of magnitude higher stability than existing organic cation-hydroxides, both in solution and under low RH at elevated temperatures. Moreover, due to the pathway of insertion and bonding mode of the internal proton, it is unlikely that its extraordinary mechanism of stabilization will be observed in different cationic structures. Herein, we present in-DBD as the only cation thus far expected to be capable of meeting lifetime requirements under the harsh conditions of vapor-phase AEM electrochemical technologies.

## Results
### Synthesis and Structural Characterization of in-DBD
In-DBD was previously reported by the Alder group[39] with the final transformations and robust chemical characterization not reported

(supplementary fig. 2), and thus redeveloped in our laboratory and adapted to present safer, more scalable methods (supplementary fig. 3). Maleic hydrazide is oxidized by lead tetraacetate, which undergoes a Diels-Alder reaction with 1,3-butadiene to produce 1,6-diazabicyclo[4.4.0]dec-3,8-diene-2,5-dione[40]. This is then subsequently hydrogenated and reduced by LiAlH$_4$ to produce 1,6-diazabicyclo[4.4.0]decane. This is singly-alkylated with 1,4-dibromobutane in THF, and then further cyclized by addition into AgBF$_4$ to produce 1,6-diazatricyclo[4.4.4.0$^{1,6}$]tetradec-1,6-diylium tetrafluoroborate[41]. The N-N bond is readily reduced by zinc in acidic media to produce the outside-protonated DBD tetrafluoroborate salt, which is then inside-protonated by oxidation with K$_2$S$_2$O$_8$ in acidic media. This inside-protonation was shown by Alder and coworkers[39] to proceed via a single-electron-transfer-mediated intramolecular H-atom abstraction of an α-proton of DBD (supplementary fig. 1), rather than direct protonation of the diamine, which implies that a proton (and thus any atom) cannot penetrate the void-space between the bridgehead nitrogen atoms. Further ion exchanges were performed using ion-exchange resin to the chloride or hydroxide form.

The caged proton of in-DBD appears in $^1$H NMR spectra at (δ) 17 ppm, evident of its extensive deshielding by the two nitrogen atoms. The crystal structure of in-DBD chloride[42] shows a very short N...N distance (2.5 Å), with a symmetrical electron density distribution between them, implying a symmetrical N-H-N bond; this hydrogen bond is shorter and more symmetrical than observed by proton sponge structures[43]. DFT calculations (ωB97XD/6−31 g(d.p)) reveal in-DBD to be 36.3 kcal/mol more stable than the outside-protonated form, which is much greater than that gained by moving a proton inside a proton-sponge (18.7 kcal/mol) (supplementary fig. 78). We stress the unique structure of this molecule, as direct inside-protonation and deprotonation is not observed, instead requiring complex radical mechanisms for inside-protonation (supplementary fig. 1) and no observed deprotonation (see below). This approach differs from typical cryptate cationic groups where the metal center is easily displaced and inserted[44]; the internal proton of in-DBD is endohedrally protected, not simply sterically protected. Collectively, these analyses support the uniqueness of the in-DBD structure, which originates from the enforced, single-well hydrogen bond and endohedral protection and the extraordinary mechanism of proton insertion.

### Assessing the unique stability of in-DBD to AEM-relevant caustic conditions
The stability of in-DBD to strongly basic conditions was studied alongside the previously described state-of-the-art cations, using two accelerated degradation methods: solution NMR spectroscopy to determine stability and degradation products and dynamic vapor sorption (DVS) to probe stability under extreme dehydration conditions encountered in AEM fuel cells, for example. Both tests are limited by practical factors. NMR studies must be carried out in solution and are not equivalent to operando conditions, as the presence of additional solvent molecules (here: MeOD) allows solvation of the molecular scaffold, especially benefitting cations that make use of steric

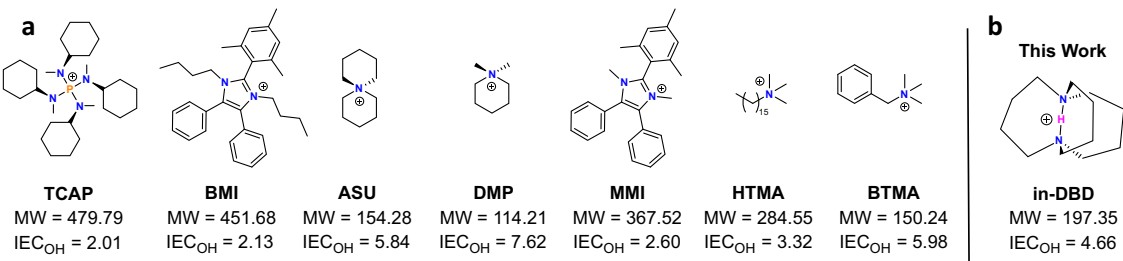

**Fig. 1 | Chemical structures of state-of-the-art organic cations. a** Typical cations used in AEM technologies, and **b** this work. IEC is reported in meq/g.

protection of their cationic centers (e.g., ASU, MMI) to prevent degradation. DVS studies provide reaction conditions similar to those in AEM-based fuel cells, with the presence of only the hydroxide counter-ion and its hydration water at a given RH; however, little mechanistic insight is obtained, and degradation products require further ex-situ characterization methods. However, the complementary nature of these characterization methods provides a uniquely holistic ex-situ look at the lifetime of cations in device-relevant caustic conditions.

## NMR degradation studies

Many common hydroxide-stability tests rely on degradation in an organic solution containing (alkali metal) hydroxide salts and certain amounts of water; however, comparability is hindered since hydration levels are rarely reported or considered despite its deciding impact on the reactivity of hydroxide. For example, it has been previously shown that MMI possesses a half-life of 7790 h in 3 M NaOD in 7:3 MeOD/$D_2O$ ($\lambda \approx 4.8$), but for much lower hydration levels ($\lambda \approx 1$) in the superbase 0.5 M [K(18-crown-6)]OH in DMSO, the half-life reduced to 92 h[23]. To accommodate the desire for low hydration conditions, we adapted a NMR spectroscopic degradation method reported by Coates et al.[45] that used 2 M KOH in $CD_3OH$ and an internal standard (sodium trimethylsilyl propane sulfonate, NaDSS) in a sealed NMR tube at 80 °C, acquiring NMR spectra at regular intervals (supplementary Tab. 1 and 2). The reported procedure was modified to maintain consistency in hydration levels ($\lambda \approx 1$) and observe exchangeable protons: solutions were prepared under argon atmosphere to limit exposure to atmospheric water and $CO_2$, and $CD_3OD$ was used instead of $CD_3OH$. It should be noted that deutero-methoxide is formed under these conditions and may cause different degradation modes than pure hydroxide degradation[20,38]. The percentage of cation remaining versus time is shown in Fig. 2a. Under these conditions, no measurable degradation of in-DBD was found over 30 days (supplementary fig. 36). Of significant note, no deuterium exchange of any proton was observed, implying no electrophilic α or β protons are present and, critically, that the inside-proton is not able to be deprotonated or exchanged for another proton. All other cations undergo degradation, characterized as a relative lowering of the cation NMR signal intensity. BTMA, used as a baseline comparison in this work, degrades rapidly under these conditions, with less than 15% BTMA remaining after 10 days and no cation detectable after 29 days. HTMA, MMI, and DMP show similar rates of degradation to each other, with DMP having slightly higher stability (54%, 65%, and 77% cation remaining after 31, 29, and 30 days, respectively). ASU reveals high stability relative to the other common cations, retaining 95% of the cation after 29 days.

[1]H NMR spectra reveal identified degradation products for all cations, except in-DBD for which no degradation products were found (supplementary fig. 38) even after 92 days (supplementary fig. 41). BTMA degrades via de-methylation and de-benzylation (supplementary fig. 23), while HTMA degrades by de-methylation and elimination (supplementary fig. 26), and MMI degrades primarily by de-methylation and secondly by hydroxide attack on C2 to give the ring-opened product (supplementary fig. 29). DMP degrades primarily by de-methylation, with a small amount by Hofmann-type elimination (supplementary fig. 32), while ASU shows exclusively Hofmann elimination (supplementary fig. 35). The half-lives of these salts in 2 M KOH in $CD_3OD$ were estimated from the slope of each degradation plot (supplementary fig. 43–48). As in-DBD exhibited no evident degradation even after over 92 days (supplementary fig. 48), its half-life is estimated at 2,15,000 h (> 24 years). In-DBD ($t_{1/2} \approx 2,15,000$ h) exhibits greater than an order of magnitude longer half-life than ASU ($t_{1/2} \approx 6950$ h), > two orders of magnitude longer than HTMA, MMI, or DMP ($t_{1/2} \approx 770$ h, $t_{1/2} \approx 990$ h, and 1440 h, respectively), and > three orders of magnitude longer than BTMA ($t_{1/2} \approx 130$ h).

## Dynamic vapor sorption degradation studies

Low hydration conditions, in the absence of a solvent, simulate extreme AEM fuel cell cathode conditions[16]. Kreuer and Jannasch[21] developed a method to study these conditions through DVS of AEMs in their hydroxide form, which we have adapted to study small molecules. Since the hydroxide anion becomes significantly more nucleophilic and basic when waters of hydration are removed[3], these studies typically reveal substantial degradation of the cationic functional groups even at modest relative humidities (60–30%), and to date, no organic cation hydroxide salt has been reported to persist below 10% RH at 80 °C[21,45]. To account for the increased nucleophilicity of hydroxide ions at low $\lambda$[3], we define critical conditions of 30% RH and 10% RH, relating to critical local reaction environments expected to occur during the operation of an AEM fuel cell at high current densities[16]. DVS gravimetric analysis of the cations in their hydroxide form at 80 °C is shown in Fig. 3 and Fig. S50–S63. Sweeping RH scans, from high to low RH conditions (Fig. 3a), reveal the RH range at which each cation begins to degrade, observed as a mass-loss over time caused by volatilization of (volatilizable) degradation products and reduced hygroscopicity (supplementary fig. 64). Figure 3b shows the pertinent stability of the measured cations; In-DBD remains stable at 10% RH, showing only minor mass loss at 8% RH at 80 °C (0.13% $h^{-1}$) and an increased rate of mass loss occurs at 5% RH (0.63% $h^{-1}$). DMP shows a higher susceptibility to degradation under reduced RH, where degradation begins as high as 23% RH (0.07% $h^{-1}$) and substantial degradation at 11% RH (0.21% $h^{-1}$). ASU begins to degrade as high as 23% RH (0.08% $h^{-1}$), with substantial degradation occurring at 17% RH (0.34% $h^{-1}$). HTMA shows relatively high initial degradation at 49% RH (0.04% $h^{-1}$), showing substantial degradation at 33% RH (0.35% $h^{-1}$).

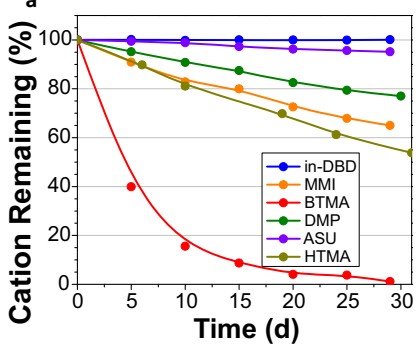

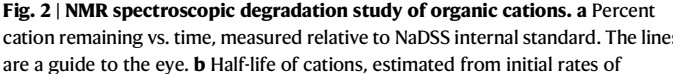

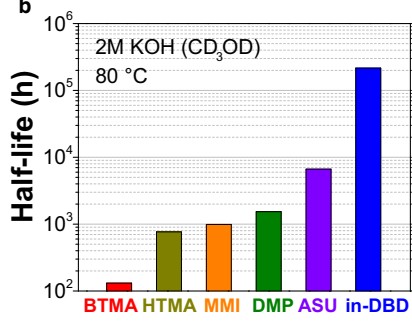

**Fig. 2 | NMR spectroscopic degradation study of organic cations. a** Percent cation remaining vs. time, measured relative to NaDSS internal standard. The lines are a guide to the eye. **b** Half-life of cations, estimated from initial rates of degradation. Degradation studies were conducted in dry $CD_3OD$ with KOH (2 M), 0.03 M analyte and 0.03 M NaDSS internal standard.

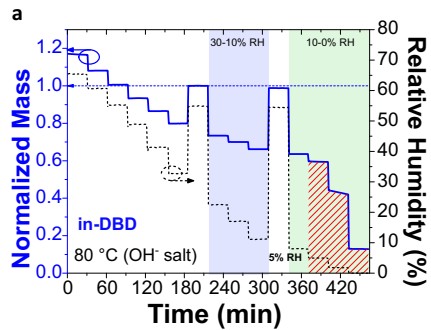

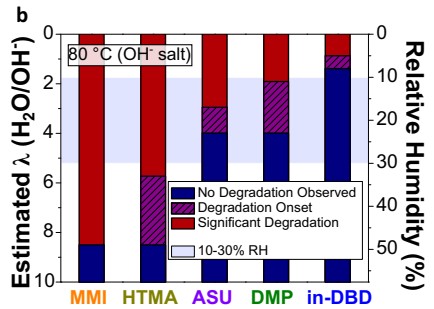

**Fig. 3 | DVS degradation studies of cation-hydroxides. a** The last 30 min of each step from the DVS step scan of in-DBD hydroxide showing the normalized mass (blue axis) and RH (black axis) over time. The blue dotted arrow represents the mass at the reference RH (55%) prior to degradation. Red shading represents the RH region where degradation occurs and the number beside is the RH where substantial degradation (>0.2% mass loss h⁻¹) is first observed. The pale blue and pale green backdrops represent 30-10 and 10-0% RH conditions, respectively. **b** RH and estimated $\lambda$ at which the studied cation-hydroxides observe a loss of mass. The dark blue represents RH where the cation has no observable mass loss (>0.04% h⁻¹), the purple hashed region represents the RH where the cation observed minor mass loss (between 0.04% and 0.2% h⁻¹), and the deep red region represents the RH where major mass loss (>0.2% h⁻¹) was observed. All DVS were obtained under a flow of argon at 80 °C. All mass was normalized to the stabilized mass at 55% RH.

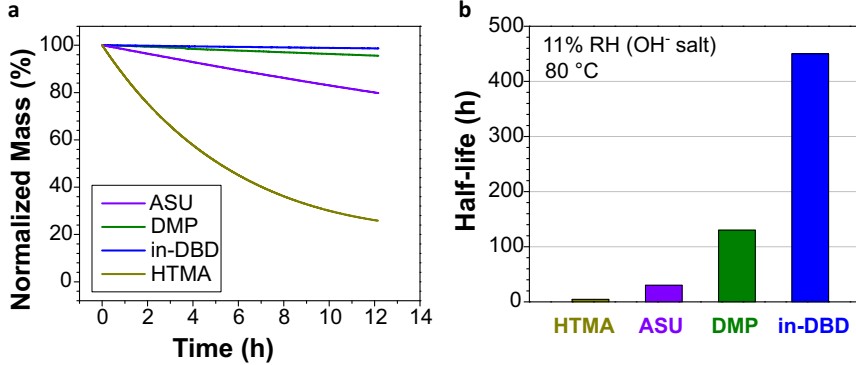

**Fig. 4 | DVS stability tests of the fully volatilized cation-hydroxides. a** The normalized mass loss over time and **b** the extracted half-life of the cations at 11% RH and 80 °C. The $\lambda$ is estimated at 2 H₂O/OH⁻.

MMI degrades relatively quickly at a modest 49% RH (0.41% h⁻¹). The relative stability of each measured cation clearly shows in-DBD is the only cation-hydroxide capable of persisting under the low-humidity conditions.

To assess the long-term stability under the high-stress condition of critically low RH, the cations in their hydroxide form were exposed to 11% RH at 80 °C for 12.2 h. Since in-DBD, DMP, ASU, and HTMA showed virtually 100% volatilization of their degradation products, their observed mass loss can be taken as a quantitative measure of degradation, where the rate of mass loss is essentially the rate of decomposition. It should be noted that a rate of decomposition cannot be collected for MMI since it had a significant amount of non-volatile decomposition products (supplementary fig. 59, 60). As seen in Fig. 4a and supplementary fig. 70–73, in-DBD hydroxide is highly stable, observing relatively little mass loss (1.3%) over 12.2 h. DMP loses more mass over the experiment (4.4%), with ASU losing significant mass (20.2%) and HTMA losing most of its mass (74.2%). The half-life of these cations is shown in Fig. 4b, where in-DBD possesses a half-life (450 h) almost 3.5 times higher than DMP (130 h), over an order of magnitude longer than ASU (30 h) and 2 orders of magnitude longer than HTMA (5 h). It is important to note that these conditions represent temporary excursions into the harshest alkaline conditions expected for AEM technologies, and normal operation would rarely reach these conditions for sizeable timespans; the extraordinarily long half-life of in-DBD under these conditions shows an unprecedented hydroxide stability relative to all other studied cations.

To determine the degradation products of in-DBD, and exclude volatilization of any unforeseen molecular species, the DVS outflow was captured in 5% v/v acetic acid solution. The only observed product was the Hofmann-type elimination product (supplementary fig. 74–76). The lack of any other degradation products, particularly no deprotonation of the caged proton, shows the incredible kinetic stability afforded by the endohedral protection and enforced hydrogen bonding. Calculations (shown below) give insight into why only one degradation product is seen, where the endohedral protection blocks hydroxide access to the central proton, and the low electrophilicity of the α and β positions require very harsh conditions before a Hofmann-type elimination is favorable.

When compared to the other cations, in-DBD is substantially more inert in both degradation testing methods and shows unprecedented stability at low (10%) RH at 80 °C. By applying these complementary degradation methods (solution NMR and vapor-phase DVS), we were also able to reflect the differences in stability that have been observed for ASU in ex-situ solution degradation studies (highly stable) versus operando polymer studies (poorly stable) and show that; conversely, in-DBD does not suffer in translating stability from solution to vapor-phase. This is owing to the unique nature of in-DBD and its one-of-a-kind stabilization concept; whereas ASU relies on maintaining the specific chair-configuration around the nitrogen atom to provide its steric protection, the endohedral protection of in-DBD ensures the hydroxide ion cannot neutralize the cation, and the intra-bridgehead hydrogen bond distributes the positive charge to lessen the

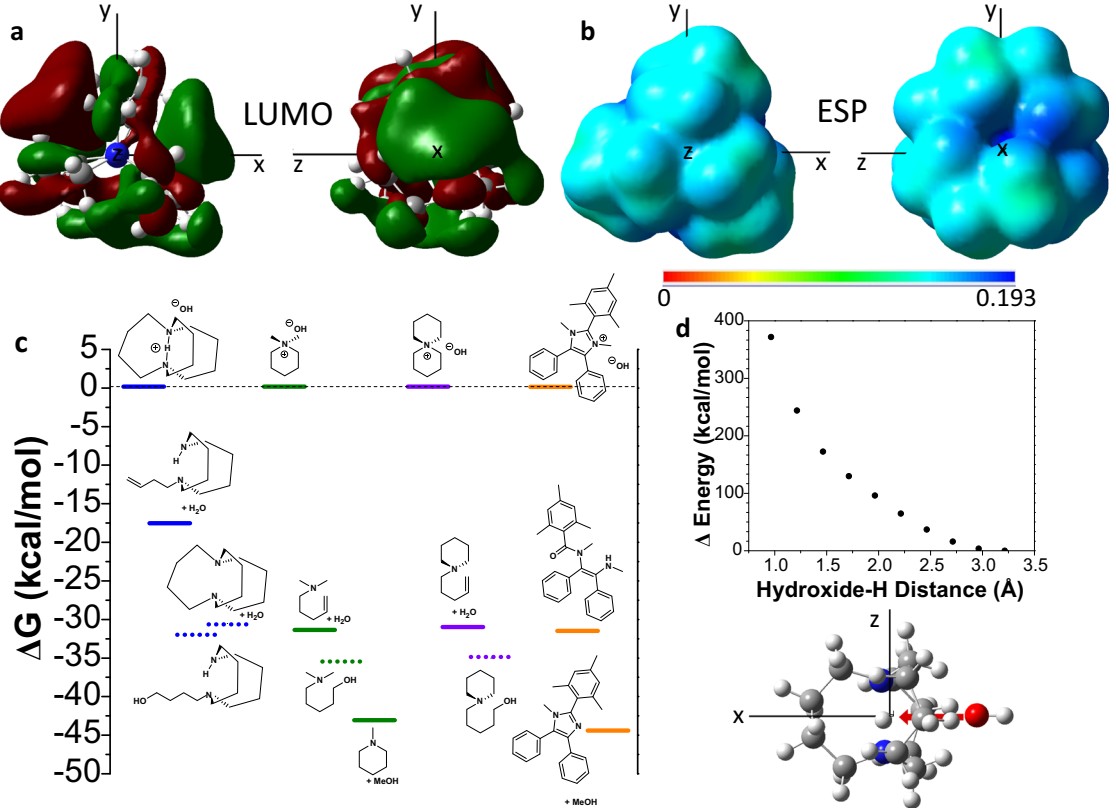

**Fig. 5 | DFT calculations of cations. a** LUMO isosurfaces and **b** electrostatic potentials (ESP) of in-DBD. ESP is shown on a scale of Hartree atomic units, where more positive numbers represent a stronger positive potential, and higher electrophilicity. **c** Free energy difference between parent cations and degradation products, calculated in a water solvent continuum model. Experimentally-unobserved degradation products are shown as dotted lines. **d** The energy barrier required to deprotonate the caged proton of in-DBD. The structure shows the approach of OH⁻ to the caged proton. DFT calculations were performed at ωB97XD/6-31 g(d.p) level of theory.

probability of elimination pathways, leading to intrinsic structural stability of in-DBD.

## DFT calculations

DFT calculations were performed at ωB97XD/6-31 g(d.p) level of theory and are found in the source data. The optimized structure for in-DBD shows the HOMO isosurface is located mainly along the N-H-N axis, with substantial electron density also located on the α-carbons next to the N-atoms (supplementary fig. 77). The LUMO isosurface shows the orbital primarily sits between the carbon rings (Fig. 5a), and there is a node at both the carbon rings and along the N-H-N axis; the two nodes present imply that hydroxide attack in these positions is highly unfavorable. The electrostatic potential (ESP) of in-DBD is shown in Fig. 5b. It shows that the void space between the alkyl rings is the region with the highest positive electrostatic potential and is thus the point where the hydroxide ion would be expected to attack; this would encourage the hydroxide to remain in the impenetrable (see below) void-space where it is likely geometrically unfavorable to undergo Hofmann-type elimination reactions and can be one of the reasons we observe such high stability to hydroxide.

We performed DFT calculations to obtain additional insight into the origin of the high hydroxide stability of in-DBD. The free energy of the degradation products of cations were calculated, and the relative energies shown in Fig. 5c. From this, it is observed that generally the degradation products are more thermodynamically stable than the parent cation-hydroxide complex. DMP and MMI both show stabilization energy of 30–45 kcal/mol for degradation products, whereas ASU shows a fairly low energy difference of ~31 kcal/mol for degradation by Hofman-type elimination and ~35 kcal/mol for the

experimentally unobserved OH-addition product. In-DBD shows elimination to be barely favorable, only being lower in free energy by ~17 kcal/mol. Deprotonated DBD is theoretically more stable by ~30 kcal/mol relative to the parent cation-hydroxide ion-pair, and the hypothetical hydroxide $S_N2$ substitution product is lower in free energy by ~32 kcal/mol. The barrier for hydroxide penetration into in-DBD leading to deprotonation was calculated (Fig. 5d) to be prohibitively large (~400 kcal/mol), confirming why this mode was not observed. Notably, although OH addition products were calculated for in-DBD, we did not observe any of these compounds. The combination of the lack of electrophilic sites, the high barrier to internal deprotonation, and the low energy difference between elimination products and parent cation serve to explain the extraordinarily high stability of in-DBD.

## Discussion

We show that **in-DBD** is immensely more resistant to hydroxide-promoted degradation than state-of-the-art organic cations, both in solution and under low RH at 80 °C. A combination of endohedral protection (cage structure) and enforced single-welled hydrogen bonding enables in-DBD hydroxide to persist at 5% RH at 80 °C. The extraordinary structure of in-DBD, where the internal proton charge carrier can neither be inserted nor deprotonated by conventional means, presents a unique finding in the search for ultra-stable, low molecular weight organic cations. The intention of this report is to stimulate the research necessary to synthetically incorporate these remarkable cations into macromolecules, fabricate dimensionally robust anion exchange membranes, and ultimately integrate them into alkaline AEM devices such as fuel cells and other vapor phase

technologies. This may help unlock fluorine-free, non-PGM electrochemical energy conversion devices for a sustainable energy future.

## Methods

All reagents were purchased from Millipore-Sigma, Fisher Scientific, or Combi-Blocks Inc., received as ACS grade or >97%, and used as received unless noted. All solvents designated as "dry" were stored over activated 4 Å molecular sieves under an argon atmosphere. KOH used in the study was obtained from Fisher Scientific, ACS grade pellets (85%, <2% $K_2CO_3$), and immediately transferred into a nitrogen glovebox upon opening, where it was powdered for use in experiments. Exchange to chloride or hydroxide ion form was performed with Amberlite IRA 900-Cl, and Amberlyst A26-OH form ion-exchange resins, respectively. Milli-Q water was obtained from a Millipore Gradient Milli-Q (Merck KGaA, Germany) water purification system with a resistance of 18.2 MΩ cm.

NMR was collected on a Bruker AVANCE III 400 MHz (routine NMR) or AVANCE III 600_QCI 600 MHz NMR (degradation experiments) at 298 K. Mass spec was collected on a Bruker maXis Impact MS, with ESI ionization. DVS was recorded on a Surface Measurements Systems DVS Adventure system with argon as the carrier gas. TGA was recorded on a Shimadzu's TGA-50 under a nitrogen atmosphere with a heating rate of 10 °C per minute. DFT calculations were performed using Gaussian 09 software suite[46] at ωB97XD/6-31 g(d.p) level of theory.

## Data availability

All processed data are available in the manuscript or the supplementary materials. Additional raw data are available from the corresponding authors upon request. Source data are provided with this paper.

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

## Acknowledgements

Natural Sciences and Engineering Research Council of Canada Discovery Grant RGPIN-2018-03698 (S.H.) Natural Sciences and Engineering Research Council of Canada Discovery Grant RGPIN-2017-04910 (A.J.B.).

## Author contributions

Conceptualization: A.J.B. and S.H. Synthesis: C.L.R. Computations: C.L.R. Methodology: C.L.R. and T.S. Investigation: C.L.R. and T.S. Visualization: C.L.R. Funding acquisition: S.H. Supervision: S.H. and A.J.B. Writing – original draft: C.L.R. Writing – review & editing: C.L.R., T.S., S.H., and A.J.B.

## Competing interests

A provisional patent application (#63555540767) has been filed by Simon Fraser University with the inventors Chase L. Radford, Torben Saatkamp, Andrew J. Bennet, and Steven Holdcroft, covering the synthesis of in-DBD and its applications. The authors declare no non-financial competing interests.
