## [Peer Review File · Nature Communications]

REVIEWER COMMENTS

Reviewer #1 (Remarks to the Author):

The authors have carefully examined the alkaline stability of a proton cage cation. The compound, abbreviated as in-DBD in the manuscript, was first described in 1978, and this proton cage is exceptionally resistant to hydroxide/methoxide-promoted degradation under low hydration levels due to the kinetically trapped endohedral proton.

In this report, the authors have an improved method for synthesizing this cationic compound as compared to the first report.

They then carry out studies of alkaline stability under low hydration, as it is known that hydroxide and alkoxide anions become more reactive with decreased hydration. This is a significant concern in electrochemical devices built using anion-exchange membrane technology where water management can lead to very low hydration and thus degradation. Studies in 2 M KOH/deuterated methanol and dynamic vapor sorption studies of the hydroxide salts under low hydration at 80 C demonstrate that this cage compound is exceptionally alkaline stable. It is the most stable reported to my knowledge. In future studies, it would be valuable to examine the stability of in-DBD relative to the tetraaminophosphonium salts, which also do not degrade in 2 M KOH/deuterated methanol at 80 C. Overall, the manuscript reports on an exciting finding for AEMs and the work is carefully executed. It should be accepted in Nature Communications after the following minor revisions.

1. References should be checked prior to publication. For example, p.3 line 11 - this sentence discusses the Alder paper and cites ref 43 but it looks like this should be ref 46 (J. Am. Chem. Soc. 101, 3652).

2. Can the authors comment on any approaches used to remove water from the deuterated methanol? Were these dried over sieves? On p.4, lines 32-45 - a statement regarding the alkaline medium would be helpful. Under the described conditions, alkaline degradation is likely driven by the deuterio-methoxide anion, which will lead to different degradation rates and potentially different degradation products than hydroxide. A statement about the anion (and a relevant reference) can be added in this section.

3. I would recommend adding a reference to the barium cryptate cations reported by Wei You and coworkers (<https://doi.org/10.1002/ange.202217742>). The in-DBD cation has some relationship to these cryptate systems.

Reviewer #2 (Remarks to the Author):

This is a great addition to the scientific literature. The "rediscovery" of this cation for AEM purposes is really interesting and will be well received by the scientific community. The work is well done and well presented and builds on the previous seemingly forgotten work of Alder et al. I think very little needs to be done to this manuscript to be published and have only 2 areas of any meaningful significance. Both involving Figure 3. The first and the only thing that I think is substantively significant is the shaded region in Figure 3b which refers to Transient Fuel Cell Conditions and has RH 10-30% RH highlighted should be removed from the Figure. While it wouldn't be impossible to force a fuel cell into this operating range and for PEM systems there has been a desire to move into this operating range, no systems are close to operating at this range as much for catalyst reasons as membrane reasons. There isn't a good justification for having this shaded region here and it isn't needed. (This also shows up in supplemental Figure 61, and while I don't care for this, it only says 10-30% RH rather than transient fuel cell conditions so it doesn't bother me as much) In Figure 3a just above the x axis at 300- 360 minutes it says 5% RH. I am unclear why this is there or what it is supposed to mean. I suggest removing. It also appears in supplemental Figure 48 and 51 and it just seems unneeded. The writing is quite good there is a minor typo in supplemental Figure 38. Percent cation remaining of in-DBD i(s) in its chloride and BF₄-ion-form.

After these issues, I have just a few additional comments not specifically critical for the manuscript but for the author's and editors benefits. First, the reason AEM fuel cells aren't viable currently isn't due to the hydroxide stability of the AEMs. It has more to do with carbonate poisoning and electrode performance and degradation. This is in part some of my concern over the highlight of a specific RH range at 10-30% where no low T fuel cells operate currently. One issue is also with catalyst - ion group interactions which could be different in these shielded ions (this is mentioned at all in the manuscript but is just as if not more interesting to be than the OH⁻ stability). Also, AEM now have more interest in electrolysis than FCs and in these cases they are fully hydrated. These comments should not take away from the novelty or interest in the results presented. Second, I was interested in the DFT results, and think they provide value, but the biggest question I have for this specific cation is how does it interact with electrocatalysts (as highlighted earlier and could be probed along the lines that other cations have been in RDE/half studies with Pt are a gap that would be good to fill) and where does the hydroxide want to reside and how easily does it conduct between different (fixed) cation positions. Along these lines conductivity values and water uptakes of salt solutions at different RH/water activity values are missing gaps that wouldn't be horribly difficult to run and would say a lot about the potential viability of these materials in functioning materials (for example in our work with P centered cations originally demonstrated by Schwesinger or Vercade we found extreme stability but correspondingly high hydrophobicity - these cations due to smaller sizes may avoid these issues but remain a potential concern for the ability to perform well in membranes. Overall, extremely interesting and well done.

Reviewer #3 (Remarks to the Author):

The manuscript by Radford et al. describes the synthesis of a cationic structure with a proton trapped within the internal cavity of inDBD, and its stability in different alkaline environments. The authors have “rediscovered” the compound for use as an alkali-stable cationic group in anion exchange membranes (AEMs) for alkaline fuel cells (AEMFCs). In this work, the synthetic pathway has been improved and the use of toxic chemicals has been reduced. Still, the presented synthesis represents a significant challenge and requires 7 steps with the use of expensive reagents, reactants and catalysts.

The most important claim in the manuscript is the exceptionally high alkaline stability of the in-DBD cation under harsh alkaline conditions. It is surprising and quite incredible that the proton in in-DBD is not neutralized under high pH conditions. The work presented is novel on a conceptual level and may very well represent a major breakthrough in the field to bring alkaline membrane fuels and water electrolyzers closer to practical applications and eventual commercialization. With the introduction of the inDBD cation, a completely new strategy towards alkali-stable AEM materials is opened up that will attract a significant interest in the field. However, the final proof of the stability and usefulness of this cations in fuel cells under low RH conditions can only be assessed after the preparation and investigations of in-DBD-functionalized AEMs. The work is of very high quality and all the conclusions are appropriately supported by data. Hence, the manuscript can be accepted in the present form.

Response to Reviewers

Reviewer #1 (Remarks to the Author):

The authors have carefully examined the alkaline stability of a proton cage cation. The compound, abbreviated as in-DBD in the manuscript, was first described in 1978, and this proton cage is exceptionally resistant to hydroxide/methoxide-promoted degradation under low hydration levels due to the kinetically trapped endohedral proton.

The authors thank the reviewers time and attention, and supportive comments.

In this report, the authors have an improved method for synthesizing this cationic compound as compared to the first report.

They then carry out studies of alkaline stability under low hydration, as it is known that hydroxide and alkoxide anions become more reactive with decreased hydration. This is a significant concern in electrochemical devices built using anion-exchange membrane technology where water management can lead to very low hydration and thus degradation. Studies in 2 M KOH/deuterated methanol and dynamic vapor sorption studies of the hydroxide salts under low hydration at 80 C demonstrate that this cage compound is exceptionally alkaline stable. It is the most stable reported to my knowledge. In future studies, it would be valuable to examine the stability of in-DBD relative to the tetraaminophosphonium salts, which also do not degrade in 2 M KOH/deuterated methanol at 80 C. Overall, the manuscript reports on an exciting finding for AEMs and the work is carefully executed. It should be accepted in Nature Communications after the following minor revisions.

In reference to tetraaminophosphonium salts, we agree in future those would be interesting points of comparison. For the present study, a selection was made to well represent the most stable cations that are also commonly employed and found in commercial AEMs of great performance and stability.

1. References should be checked prior to publication. For example, p.3 line 11 - this sentence discusses the Alder paper and cites ref 43 but it looks like this should be ref 46 (J. Am. Chem. Soc. 101, 3652).

This was an update error from the reference managing software. The authors thank the reviewer for catching this mistake, it has been corrected. After updating the reference list it is ref 39.

2. Can the authors comment on any approaches used to remove water from the deuterated methanol? Were these dried over sieves? On p.4, lines 32-45 - a statement regarding the alkaline medium would be helpful. Under the described conditions, alkaline degradation is likely driven by the deuterio-methoxide anion, which will lead to different degradation rates and potentially different degradation products than hydroxide. A statement about the anion (and a relevant reference) can be added in this section.

As noted in the SI (S16) "NMR samples were prepared under argon to limit the water present in the sample to that of the native water present in the KOH pellets (85% m/m, $\lambda \approx 0.55$) and commercial CD3OD (99%)...". Commercial CD3OD (Milipore Sigma) is shipped at 99% purity, with <1% water by mass. No additional drying step was taken, apart from handling the bottle under argon atmosphere directly after receipt, since the water content from the native KOH pellets and commercial CD3OD present conditions that are relatively simple and consistent for other labs to replicate and provide a $\lambda \approx 1$.

A statement is added about the formation of deuterio-methoxide: “It should be noted that deuterio-methoxide is formed under these conditions and may cause different degradation modes than pure hydroxide degradation.” And references to the original study designed by the Coates group (10.1021/acsenergylett.9b00908, and 10.1021/acs.joc.0c02051) are added.

3.I would recommend adding a reference to the barium cryptate cations reported by Wei You and coworkers (<https://doi.org/10.1002/ange.202217742>). The in-DBD cation has some relationship to these cryptate systems.

We agree that our work is relevant, but different, to this, and have added it to page 4, line 15: “This approach differs from typical cryptate cationic groups where the metal center is easily displaced and inserted; (10.1002/ange.202217742) the internal proton of in-DBD is endohedrally protected, not simply sterically protected.”

Reviewer #2 (Remarks to the Author):

This is a great addition to the scientific literature. The "rediscovery" of this cation for AEM purposes is really interesting and will be well received by the scientific community. The work is well done and well presented and builds on the previous seemingly forgotten work of Alder et al.

The authors thank the reviewer's time and attention, and the supportive comments.

I think very little needs to be done to this manuscript to be published and have only 2 areas of any meaningful significance. Both involving Figure 3. The first and the only thing that I think is substantively significant is the shaded region in Figure 3b which refers to Transient Fuel Cell Conditions and has RH 10-30% RH highlighted should be removed from the Figure. While it wouldn't be impossible to force a fuel cell into this operating range and for PEM systems there has been a desire to move into this operating range, no systems are close to operating at this range as much for catalyst reasons as membrane reasons. There isn't a good justification for having this shaded region here and it isn't needed. (This also shows up in supplemental Figure 61, and while I don't care for this, it only says 10-30% RH rather than transient fuel cell conditions so it doesn't bother me as much)

We agree with the reviewer's assessment of *typical* fuel cell operations. These conditions are labeled as "transient fuel cell conditions" for exactly the reasons mentioned by the reviewer. It is not ideal to operate under these low RH conditions, however studies by Dekel (10.1016/j.jpowsour.2017.07.012) and León (10.1016/j.apenergy.2022.119722) demonstrate that the consumption of water within the catalyst layer, coupled with electro-osmotic drag can leave the cathode of an AEM fuel cell within this range, and local regions near a catalytic center can reach a λ of 1-2 (roughly equivalent to the 10% RH value we state as the minimum). While we agree that no engineer would *like* a fuel cell to operate in these conditions, and typically the bulk RH remains above these RH values, the transient nature of these low RH conditions coupled with the rapid degradation of typical AEM headgroups means that even if the bulk is above these RH, we still need to design material that is stable within these low hydration regions. As a concession, we have changed the label to simply read 10-30% RH, as per the label in the SI. In our view the region also provides a visual indicator for researchers that if a cation degrades before entering this region, then it is unsuitable for fuel cells and other low-hydration technologies.

In Figure 3a just above the x axis at 300- 360 minutes it says 5% RH. I am unclear why this is there or what it is supposed to mean. I suggest removing. It also appears in supplemental Figure 48 and 51 and it just seems unneeded.

In the figure caption it is defined "Red shading represents the RH region where degradation occurs, and the number beside is the RH where substantial degradation (>0.2% mass loss h⁻¹) is first observed." The 5% RH is where we observe a major mass loss rate and is labeled for clarity. With due deference, we prefer to keep the "5% label" as some might find it helpful.

The writing is quite good there is a minor typo in supplemental Figure 38. Percent cation remaining of in-DBD i(s)n its chloride and BF₄⁻ ion-form.

The authors thank the reviewer for catching this error. It has been corrected.

After these issues, I have just a few additional comments not specifically critical for the manuscript but for the author's and editors benefits.

While not directly affecting the current manuscript, we thank the reviewer's following comments and points of discussion which will help frame and refine our future work within this area.

First, the reason AEM fuel cells aren't viable currently isn't due to the hydroxide stability of the AEMs. It has more to do with carbonate poisoning and electrode performance and degradation.

We agree that there are additional concerns with AEM fuel cell development, and do not mean to imply that this work alone will completely solve AEM technology. We use the phrase "unlocking vapor phase technology" to allude to this point; by making the AEM group stable to hydroxide, we can begin to decouple membrane degradation from catalyst performance and degradation.

This is in part some of my concern over the highlight of a specific RH range at 10-30% where no low T fuel cells operate currently.

As mentioned above, we agree that AEM fuel cells are unlikely be intended to operate in the 10-30% RH range (meaning 10-30% inlet gas humidity). However, at high current density, due to water consumption at the cathode, the localized RH can be significantly lower than the inlet RH, and is reported to fall within the 10-30% RH range.

One issue is also with catalyst - ion group interactions which could be different in these shielded ions (this is mentioned at all in the manuscript but is just as if not more interesting to be than the OH⁻ stability).

The authors agree that the interaction of ionic head groups with a catalyst surface is important and interesting, and is being discussed increasingly by the community. A cation that is ultra stable to OH⁻, such as the one presented here, will surely facilitate such studies, which are underway but well beyond the scope of the present report.

Also, AEM now have more interest in electrolysis than FCs and in these cases they are fully hydrated. These comments should not take away from the novelty or interest in the results presented.

This is true, we agree that current AEM technology is currently better suited for electrolysis. However, lack of a hydroxide-stable cation has hindered AEM fuel cell development. Moreover, **in-DBD** will allow AEM technology to operate in lower humidity conditions and higher temperatures than previously possible, accessing higher temperature AEM water electrolysis and other vapour phase AEM technologies.

Second, I was interested in the DFT results, and think they provide value, but the biggest question I have for this specific cation is how does it interact with electrocatalysts (as highlighted earlier and could be probed along the lines that other cations have been in RDE/half studies with Pt are a gap that would be good to fill) and where does the hydroxide want to reside and how easily does it conduct between different (fixed) cation positions. Along these lines conductivity values and water uptakes of salt solutions at different RH/water activity values are missing gaps that wouldn't be horribly difficult to run and would say a lot about the potential viability of these materials in functioning materials (for example in our work with P centered cations originally demonstrated by Schwesinger or Vercade we found extreme stability but correspondingly high hydrophobicity - these cations due to smaller sizes may avoid these issues but remain a potential concern for the ability to perform well in membranes. Overall, extremely interesting and well done.

We thank the reviewer for sharing their thoughts on the interesting aspects raised in their comment. These are all excellent suggestions for future work and we agree that these are valuable studies. We are actively pursuing more in-depth computational and experimental studies into in-DBD, as well as their integration into polymers. We welcome further discussions on areas of interest and further studies.

Reviewer #3 (Remarks to the Author):

The manuscript by Radford et al. describes the synthesis of a cationic structure with a proton trapped within the internal cavity of inDBD, and its stability in different alkaline environments. The authors have “rediscovered” the compound for use as an alkali-stable cationic group in anion exchange membranes (AEMs) for alkaline fuel cells (AEMFCs). In this work, the synthetic pathway has been improved and the use of toxic chemicals has been reduced. Still, the presented synthesis represents a significant challenge and requires 7 steps with the use of expensive reagents, reactants and catalysts.

The most important claim in the manuscript is the exceptionally high alkaline stability of the in-DBD cation under harsh alkaline conditions. It is surprising and quite incredible that the proton in in-DBD is not neutralized under high pH conditions. The work presented is novel on a conceptual level and may very well represent a major breakthrough in the field to bring alkaline membrane fuels and water electrolyzers closer to practical applications and eventual commercialization. With the introduction of the inDBD cation, a completely new strategy towards alkali-stable AEM materials is opened up that will attract a significant interest in the field. However, the final proof of the stability and usefulness of this cations in fuel cells under low RH conditions can only be assessed after the preparation and investigations of in-DBD-functionalized AEMs. The work is of very high quality and all the conclusions are appropriately supported by data. Hence, the manuscript can be accepted in the present form.

We thank the reviewer for their time and attention, and their supportive comments. We agree that the incorporation of in-DBD into AEMs is ultimately what will decide its technological impact, however we felt it would be a disservice to withhold the discovery for the years it would take to fully realize the incorporation, membrane formation, and device performance. By making this work public at its early conception, we hope to stimulate accelerated global research into this molecule so as to rapidly advance electrochemical energy conversion.